# Warming accelerates the decomposition of root-derived hydrolysable lipids in a temperate forest and is depth- and compound class-dependent

**Binyan Sun[1], Cyrill Zosso[1,3], Guido L. B. Wiesenberg[1],**

**Elaine Pegoraro[2], Margaret S. Torn[2], and Michael W. I. Schmidt[1]**

[1]University of Zurich, Department of Geography, Zurich, Switzerland.

[2]Climate and Ecosystem Sciences Division, Lawrence Berkeley National Laboratory, Berkeley, CA, USA.

[3]Agroscope, Switzerland

*Correspondence to*: Binyan Sun, binyan.sun@geo.uzh.ch

**Abstract.**

Global warming could potentially increase the decomposition rate of soil organic matter (SOM), not only in the topsoil (< 20 cm) but also in the subsoil (> 20 cm). Despite its low carbon content, subsoil holds on average nearly as much SOM as topsoil across various ecosystems. However, significant uncertainties remain regarding the impact of warming on SOM decomposition in subsoil, particularly root-derived carbon, which serves as the primary organic input at these horizons. In a whole-soil field warming experiment at Blodgett Forest Research Station (California, USA), we investigated whether warming accelerates the decomposition of root-derived hydrolysable lipids in the top- (10-14 cm) and subsoil (45-49, 85-89 cm) by using molecular markers and *in-situ* incubation of $^{13}$C-labeled root litter at each depth. Our results reveal that at compound-class level, hydrolysable lipids presented compound-dependent responses. Warming consistently reduced fatty acid mass change across soil depths, particularly at 85-89 cm. In subsoil, there was accumulation of fatty acids, which primarily originated from microbial-derived mid-chain fatty acids such as octadecanoic acid ($C_{18:0}$ fatty acids), octadecenoic acid ($C_{18:1}$ fatty acids), and hexadecanoic acid ($C_{16:0}$ fatty acids). Higher temperature attenuated this accumulation, indicating less microbial transformation of root-derived carbon under warming. At monomer level, $\omega$-hydroxy acids and diacids as suberin markers were more resistant to decomposition than bulk root-derived carbon and their resistance increased with chain-length. Moreover, warming accelerated decomposition of

individual suberin monomers in the topsoil but suppressed it in the subsoil. The slower decomposition in the subsoil was likely due to lower microbial abundance and lower soil moisture induced by warming. Our study demonstrates that the impact of warming on the decomposition of root-derived hydrolysable lipids in a temperate forest is compound class- and depth-dependent. The persistence of long-chain $\omega$-hydroxy acids and diacids may provide a potential way for long-term carbon stabilization in subsoil under climate change. Nevertheless, due to the substantial heterogeneity of subsoil environment, further studies are required to confirm and generalize this finding.

**Keywords**

Soil warming, decomposition, subsoil, hydrolysable lipids, suberin, fine roots, priming

**Highlights**

Warming effect on decomposition of root-derived hydrolysable lipids was compound class-dependent

Warming attenuated the accumulation of fatty acids in the subsoil

Suberin markers are more resistant to decomposition and their resistance increased with chain-length

Warming accelerated decomposition of suberin-derived monomers but inhibited it in the subsoil

No priming effects on the pre-existing bulk soil carbon and hydrolysable lipids after 3 years

## 1. Introduction

Global air temperatures are projected to increase between 2.6°C and 4.8°C by 2100 under Representative Concentration Pathway 8.5, according to the Intergovernmental Panel on Climate Change (IPCC, 2013). In synchrony with air temperature, soil temperature is also expected to increase, not only in topsoil (< 20 cm) but also in subsoil (> 20 cm) (Soong et al., 2020). Global soils hold the largest actively cycling terrestrial carbon pool and store between 2000 Pg and 3000 Pg of carbon in the top 3 m, with over 50% located in subsoil (Scharlemann et al., 2014). Previous studies demonstrated that warming could accelerate the decomposition of soil organic matter (SOM) in topsoil (Scharlemann et al., 2014), as well as in subsoil (Hicks Pries et al., 2017; Soong et al., 2021), potentially causing loss of $CO_2$ to the atmosphere. Moreover, enhanced temperature accelerated the decomposition of complex polymeric organic matter (Ofiti et al., 2023; Zosso et al., 2023), which had been regarded as comparatively recalcitrant to microbial decomposition.

Often, manipulative field warming experiments have focused on topsoil (Chen et al., 2022; van Gestel et al., 2018; Melillo et al., 2017; Verbrigghe et al., 2022), both in terms of the soil depths warmed by the manipulation and focus of the investigation. Therefore, it remains unclear from these experiments if the deeper soil horizons respond in similar ways to environmental changes as topsoil, since biotic and abiotic properties differ. Subsoil SOM has been assumed to be relatively insensitive to warming, because larger proportions of this deep SOM are more spatially inaccessible to microorganisms due to their associations to mineral surfaces compared to surface soil (Lützow et al., 2006). Furthermore, the microbial abundance varies throughout the soil profile (Rumpel et al., 2012; Zosso et al., 2021). Microbial biomass is substantially higher in topsoil than in subsoil (Naylor et al., 2022), by as much as two orders of magnitude (Fierer et al., 2003), leading to significantly slower turnover of carbon in the subsoil (Spohn et al., 2016). Also, microbial community structure changes with depth. Across different ecosystems, there is generally a proportional increase of Gram-positive to Gram-negative bacteria with depth (Eilers et al., 2012; Xu et al., 2021; Zosso et al., 2021) due to decreased carbon availability and quality (Fanin et al., 2019; Naylor et al., 2022). With warming, the difference in microbial abundance and composition between topsoil and subsoil could become more pronounced (Fontaine et al., 2007; Zosso et al., 2021), leaving large uncertainties of the impact on SOM decomposition at different depths.

One of the most important biotic factors that could change the SOM dynamics in the
subsoil is root mass. Compared to topsoil, root mass is one of the major carbon sources in the
subsoil (Button et al., 2022; Rumpel and Kögel-Knabner, 2011), especially in seasonally dried
temperate evergreen forest, where root depth could be deep (Schenk and Jackson, 2005). The
dual role of roots can affect SOM dynamics in subsoil in contrasting ways. On one hand, root-
derived carbon is more likely than aboveground plant biomass to associate with minerals
(Jackson et al., 2017; Rasse et al., 2005; Sokol and Bradford, 2019), thus promote carbon
stabilization. On the other hand, they can also stimulate microbial mineralization of previously
stabilized SOM, leading to loss of old SOM in the subsoil (Dijkstra et al., 2021; Fontaine et al.,
2007). However, it remains uncertain how root biomass and its microbial decomposition will
respond to warming, particularly in subsoil. Previous studies showed a variety of responses of
root biomass in the surface soil to warming, with either more fine root biomass (Kwatcho
Kengdo et al., 2022; Malhotra et al., 2020; Wang et al., 2021), less root biomass (Arndal et al.,
2018; Ofiti et al., 2021), or no change in root biomass (Wang et al., 2017). In subsoil, it is
assumed that roots will forage in deeper soil horizons under water stress induced by warming,
(Wang et al., 2017; Wang et al., 2021) but the opposite was observed in a warming experiment
in a temperate forest at Blodgett forest with a substantial loss of fine (< 2 mm) and coarse roots
(2-5 mm; Ofiti et al., 2021) across the soil profile after 4 years of warming. Additionally, it
was also observed that warming (Parts et al., 2019; Yaffar et al., 2021) or higher temperature
induced drought (Meier and Leuschner, 2008) increased mortality of fine root biomass. Since
this root litter could serve as new substrates for carbon-limited subsoil horizons and fuel
decomposition of pre-existing carbon, it is important to understand how microorganisms will
respond to this new input at different depths under warming conditions.
Although previous studies have provided evidence of carbon loss under warming at
Blodgett Forest (Soong et al., 2021), mainly by rapid decomposition of decadal-aged carbon
(Hicks Pries et al., 2017), they have not addressed the transformation of new carbon inputs into
SOM formation at the molecular level. Molecular proxies or markers such as plant-derived
hydrolysable lipids suberin, which mainly derive from woody tissues such as roots
(Kolattukudy, 1980), could be used as quantitative and qualitative methods to follow alterations
of root-derived carbon during decomposition and determine their turnover rate when in
combination with compound-specific $^{13}C$ isotopic analysis (Feng et al., 2010). Moreover, it is
important to know whether these molecular proxies could be preserved under warming.
Because applying genetic modifications to improve plant traits associated with carbon
sequestration (defined as "harnessing roots"), specifically to increase their hydrolysable lipids
content, is regarded as one of the solutions to mitigate climate change (Eckardt et al., 2023).
Many previous studies on the mechanisms of the interaction between root-derived
carbon and soil (decoupling carbon from mineral protection or formation of soil carbon) were
conducted as laboratory incubation experiments (Keiluweit et al., 2015; Sokol and Bradford,
2019). Such techniques with high replicability and controlled conditions contribute to
understanding certain soil carbon transformation or stabilization processes. However,
laboratory incubations are usually far from natural conditions and lack many of the biotic or
abiotic interactions that occur in soil *in-situ*.
Therefore, in our study, we used a long-term, whole-soil warming experiment, located
at Blodgett Forest Research Station (CA, USA), to study the effect of warming on the
decomposition of root litter at different soil depths (10-14, 45-49, and 85-89 cm). We incubated
$^{13}$C-labeled roots *in-situ* for three years, to understand how the decomposition of root-derived
hydrolysable lipids varies with depth and warming. Specifically, we quantified different
monomers in hydrolysable lipids that were released from polymeric SOM. For each monomer
we investigated stable carbon isotope values ($\delta^{13}$C) to understand how quickly root-derived
carbon and molecular markers degraded at different soil depths with warming. We
hypothesized that: first, warming would accelerate the decomposition of root-derived
hydrolysable lipids throughout the entire soil profile, regardless of soil depths. Second,
hydrolysable lipids would accumulate relative to bulk root-derived carbon due to their greater
chemical persistence compared to other carbon compounds (Lorenz et al., 2007).
**2. Material and methods**
**2.1 Study site**
The whole-soil warming experiment at University of California Blodgett Experimental
Forest is located on the foothills of the Sierra Nevada near Georgetown, CA (120°39′40″W;
38°54′43″N) at 1370 m above sea level (Hicks Pries et al., 2018). The site has a Mediterranean
climate with a mean annual air temperature of 12.5°C and a mean annual precipitation of 1774
mm (Bird and Torn, 2006). The experiment is situated in a thinned 80-year-old mixed
coniferous temperate forest, dominated by ponderosa pine (*Pinus ponderosa*), sugar pine
(*Pinus lambertiana*), incense cedar (*Calodefrus decurrens*), white fir (*Abies concolor*), and
douglas fir (*Pseudotsuga menziesii*; Hicks Pries et al., 2017). The soils are Holland series and
classified as fine-loamy, mixed superactive, mesic ultic Alfisol of granitic origin (mean pH
5.5), which is equivalent to Dystric Cambisols (IUSS Working Group WRB, 2022).
Briefly, the whole-soil warming experiment consists of 6 plots in total arranged in three
replicated blocks, each having a pair of warmed and controlled circular plots 3 m in diameter.
To maintain warming down to 1 m depth, twenty-two 2.4-m-long resistance heating cables
(BriskHeat, Ohio, USA) were vertically installed in metal conduits at a radius of 1.75 m,
surrounding each plot. Two concentric rings of surface heater cable were installed at 1 and 2
m in diameter, 5 cm below the soil surface, to compensate for surface heat loss. The average
soil temperature was elevated by + 4°C in warmed plots relative to ambient plots (except for at
0-20 cm only elevated by +2.6°C due to surface heat loss), while preserving seasonality and
natural temperature gradient with soil depth (Hicks Pries et al., 2017; Pegoraro et al., 2025; in
review). The setup of the control plots is identical to the warmed plots but without heating
cables placed inside the metal conduits (Hicks Pries et al., 2017).
**2.2 $^{13}$C-labeled root litter experiment and sampling**
Common wild oat (*Avena fatua)* is an annual grass whose fast growth enabled uniform
$^{13}$C-labelling in this experiment compared in contrast to the slower-growing native coniferous
trees. This made its roots a suitable model substrate for decomposition studies (Hicks Pries et
al., 2018). *Avena fatua* seedlings were grown for 12 weeks in a greenhouse within an airtight
chamber at the University of California, Berkeley. Every 4 days the source of $CO_2$ was
switched between ambient $CO_2$ and 10 atom% $^{13}CO_2$ (Cambridge Isotope Laboratories, Inc.,
Massachusetts, USA). After this labeling phase, roots were excavated, dried and cut in 1-2 cm
pieces (< 2 mm diameter). The root substrate was enriched by 5.6% atom% $^{13}$C and had a
carbon concentration of 0.463 g C g root$^{-1}$ (Castanha et al., 2018; Hicks Pries et al., 2018).
A total of six soil cores were extracted in each plot by using a perforated custom coring
system made of polycarbonate tube (5.04 cm outer diameter and 4.41 cm inner diameter). Each
tube consisted of four sections (10 cm, 35 cm, 40 cm, and 10 cm in length, respectively) that
were threaded on each end (male threads) and connected with female-to-female threaded
polycarbonate connectors. Each section was threaded onto a sharpened aluminum cutting edge
at the bottom for coring, and an aluminum tube with a weighted pounding head on the top. We
cored the mineral soil based on the length of each polycarbonate section at the following depths:
0-10, 10-45, 45-85, and 85-95 cm. The top 4 cm of soil in each section (target experiment depth:
10-14, 45-49, and 85-89 cm) was marked and scooped into aluminum tins. Then a pre-weighed,

0.14 g aliquot of *Avena fatua* fine roots were added. The labeled *Avena* root litter was added to four out of the six cores and these cores will be termed as *root litter* treatment in this paper. The same disturbance was applied to the rest of the two cores without addition of labeled root litter and will be denoted as *disturbance control* treatment. Mesh disks with 1mm aperture were positioned at the top and bottom of each target 4 cm soil increment to delineate the physical boundaries of the inserted root substrates. This setup was intended to retain the added root litter and prevent ingrowth of living roots into the cores and was also done in disturbance control treatment. The soil mixture was then added back to each polycarbonate section. The polycarbonate core sections were consequently connected with the female-to-female connectors and the resulting 95 cm long core was placed back in the soil. In July 2019, i.e., after three years of *in-situ* field incubation, two cores with root litter treatment and one core with disturbance control treatment were retrieved from each plot. The retrieved cores were wrapped in foil and transported to Lawrence Berkeley National Laboratory and stored in a -20°C freezer before further processing. The more detailed description of experimental setup and incubation methods is demonstrated in Pegoraro et al. (2025; in review). For brevity, DC will be used to refer to the disturbance control treatment in the rest of this paper.

Because prior work at this site concentrated on three depths (Hicks Pries et al., 2017; Soong et al., 2021), we selected the same depths to facilitate cross-study, temporal and spatial comparisons. These depths also capture major pedogenic zones: topsoil, a transitional mid-depth between topsoil and subsoil, and the deep subsoil horizon.

## 2.3 Soil preparation and characterization

In the results and discussion for this paper, the topsoil denotes the soil depth at surface (10-14 cm), and the subsoil describes the soil depths at 45-49 cm (mid-depth) and 85-89 cm (deep soil).

In the laboratory, we opened the polycarbonate cores to retrieve the 4 cm section where labeled roots were added (same for disturbance control). We also sampled the 4 cm section above and below the target depths. The soil samples were sieved < 2 mm. Roots were picked off the top of the sieve by tweezers. Bulk soil < 2 mm was freeze-dried and re-weighed. A subsample of bulk soil samples was ground by a ball mill (MM400, Retsch, Haan, Germany) and analyzed for carbon and nitrogen concentrations, as well as stable carbon isotope composition ($\delta^{13}$C) using an elemental analyzer-isotope ratio mass spectrometer (EA-IRMS;

Flash 2000-HT Plus, linked by Conflo IV to Delta V Plus isotope ratio mass spectrometer, Thermo Fisher Scientific, Bremen, Germany). The results are reported in the $\delta$ notation:

$$\delta^{13}C = \left(\frac{R_{sample}}{R_{standard}} - 1\right) \times 1000 \qquad (1)$$

Where $R_{sample}$ and $R_{standard}$ are the $^{13}C/^{12}C$ ratios of the sample and the international standard, Vienna Pee Dee Belemnite (VPDB, 0.01118), respectively. At least two analytical replicates were measured for all samples. Calibration was carried out using IAEA-certified primary standards (e.g., N600 caffeine) and caffeine (Merck) as secondary standard.

**2.4 Analysis of hydrolysable lipids**

All soil samples (< 2 mm) were pre-extracted by Soxhlet following an established protocol (Wiesenberg and Gocke, 2017) to remove solvent-extractable lipids with dichloromethane : methanol (93:7; v/v) for 48 hours. The extraction residues were dried until constant weight.

The extraction residues were homogenized with a ball mill (MM400, Retsch, Haan, Germany) and then hydrolyzed according to Zosso et al. (2023). Therefore, an aliquot of each residue equivalent to > 20 mg carbon was weighed in a 250 mL round bottom flask. The sample was mixed with the extraction solution methanol: deionized water (9:1; v/v) with 6% potassium hydroxide (KOH) and then saponified for 20 hours at around 85-88°C in a water bath under reflux. Subsequently, the solution was filtered and transferred to a separation funnel for phase separation. The solution was acidified to pH 2.0 using 6 M hydrochloric acid (HCl) and then extracted with dichloromethane. The collected fractions were volume-reduced and remaining water was removed by water-free sodium sulfate ($Na_2SO_4$).

For quantification of hydrolysable lipids, deuterated eicosanoic acid ($D_{39}C_{20}$; Cambridge Isotope Laboratories, Inc.) was added to the samples as an internal standard. The samples were silylated at 80°C for 1 hour with bis(trimethylsilyl)acetamide (BSA; Wiesenberg and Gocke, 2017). Individual compounds were quantified on an Agilent 7890B gas chromatograph (GC) equipped with a multi-mode inlet and a flame ionization detector (FID). Compound identification was performed on an Agilent 6890N GC equipped with split/splitless inlet and coupled to an Agilent 5973 mass selective detector (MS). Compounds were identified by comparison of mass spectra with those of external standards and from the NIST and Wiley mass spectra library. Both instruments were equipped with DB-5MS columns (50 m × 0.2 mm × 0.33 μm) and 1.5 m de-activated pre-columns, with helium as the carrier gas (1 ml min$^{-1}$). Silylated fractions were injected in splitless mode at an initial GC oven temperature of 50°C

that was kept isothermal for 4 min, then increased to 150°C at a rate of 4°C min[-1]. Thereafter, the temperature ramped up to 320°C at a rate of 3°C min[-1] and held for 40 min. The GC-MS was operated in electron ionization mode at 70 eV and scanned from m/z 60-650. The analysis of the data was processed with Agilent Chemstation software. The concentration of each compound was finally normalized to the organic carbon concentration of the respective sample (stated as μg g[-1] OC). The weight of samples weighed in for hydrolyzation is always corrected by accounting for the mass loss due to free lipid extraction:

$$M_{corrected} = \frac{M_{weighed}}{(1 - p_{free\ lipids})}$$ (2)

Where $M_{corrected}$ is the corrected weight of soil samples, $M_{weighed}$ is the weight of soil samples weighed in for hydrolyzation, and $p_{free\ lipids}$ is the proportion of free extractable lipids to the mass of soil samples weighed in for Soxhlet extraction.

In this paper, mid-length and long-chain monomers are the compounds with a carbon chain length $n$ between 14 and 20 ($14 \leq n < 20$) and length $\geq 20$, respectively. For each compound class, fatty acids are the synonymously used for $n$-carboxylic acids , $\omega$-hydroxy acids are short for $\omega$-hydroxy carboxylic acids, diacids stand for $a, \omega$-alkanedioic acids, alcohols are abbreviated for $n$-alcohols, and mid-chain acids stand for mid-chain hydroxylated fatty acids, referring to fatty acids with functional groups or structural modifications located in the middle of their carbon chain, typically at the C-9 and C-10 carbon positions, such as x, $\omega$-dihydroxyhexadecanoic acid (x = 9 or 10; Graça, 2015).

Different monomers can be used as markers for leaf and needle (cutin) or woody and root (suberin) biomass. However, there are no universal markers across a variety of studies since the relative proportions of cutin and suberin markers could vary among plant taxa, plant functional type or plant organ (Jansen and Wiesenberg, 2017; Mueller et al., 2012). Here, we selected $\omega$-hydroxy alkanoic acids and diacids as suberin markers since these monomers exist in substantial amount in the roots of the dominating plant species around the experiment plots and are 10 times higher in concentration compared to the same monomers in leaves or needles in the same species (Supplementary *Fig. S2*). Cutin markers could not be distinguished since all the mid-chain acids which were traditionally considered as cutin markers were present in considerable amounts in added [13]C-labeled roots (Supplementary *Table S1*).

To the best of our knowledge, there are no studies reporting the composition of mid-chain fatty acids, $\omega$-hydroxy or diacids from microorganisms in soil, although it was reported that microorganisms can synthesize the compound classes mentioned above (Huf et al., 2011; Kim and Park, 2019). These compound classes, however, are more regio-specific (the enzymes

will target only one or a few defined carbon positions) and differ from those in plants and
animals (Kim and Oh, 2013) and usually have a carbon chain-length < 20 (Zhang et al., 2024).
Therefore, we hypothesize that all the compounds with a carbon chain length ≥ 20 and mid-
chain fatty acids are exclusively plant-derived.
**2.5 Compound-specific isotope analysis**
To determine the $\delta^{13}C$ of individual compounds, a Trace GC Ultra, coupled via GC
Isolink II and Conflo IV to Delta V Plus isotope mass spectrometer (Thermo Fisher Scientific)
was used to perform compound-specific $\delta^{13}C$ analysis of individual hydrolysable lipids. The
settings of the instrument and temperature program used here was the same as mentioned above.
Reproducibility and stability (<0.6‰) of $\delta^{13}C$ values were checked with pulses of $CO_2$
reference gas and *n*-alkane standard mixture ($C_{20-30}$; Sigma Aldrich) of known isotope
composition. The $\delta^{13}C$ values were presented in per mil (‰) relative to the Vienna-Pee Dee
Belemnite (V-PDB) reference standard. Every sample was measured with three analytical
replicates and the difference between measurements typically did not exceed 1.0 ‰ for natural
abundance samples and 10% of the measured isotope value for $^{13}C$ labeled samples.
**2.6 Calculations**
The isotope composition of individual hydrolysable lipids was corrected for the value
of the $\delta^{13}C$ value of each trimethylsilyl group that was added during silylation as:
$$\delta_{UD} = \frac{(n+3\times a)\times\delta_D - 3\times a\times\delta_M}{n} \tag{3}$$

Where *n* is the number of carbon atoms in the underivatized hydrolysable lipids and
$\delta_{UD}$ and $\delta_D$ are isotope ratios of the underivatized and the derivatized hydrolysable lipids,
respectively, a is the number of functional groups in individual compounds that were
derivatized by BSA. $\delta_M$ is the carbon isotope ratio of the added trimethylsilyl group (-44.3‰).
$\delta_M$ was determined by repeated measurement (n = 8, with 3 analytical replicates of each) of
derivatized standard FAs ($C_{10}$ and $C_{12}$ FAs with known $\delta^{13}C$ isotope composition).
The $^{13}C$-excess, which can be expressed as percent atom excess, presents the
enrichment of $^{13}C$ in individual hydrolysable lipids. The value is defined as the ratio of the
relative abundance of the heavier stable isotope in a labeled sample to the natural isotope
abundance in the identical unlabeled sample (Epron et al., 2012). It was calculated as
followings (Speckert et al., 2023):

$$^{13}C - excess[\%] = \left( \frac{100}{^{13}C/^{12}C_{distcontrol}} \times {}^{13}C/{}^{12}C_{labelled} \right) - 100 \tag{4}$$

Where $^{13}C/^{12}C_{distcontrol}$ is the atomic ratio of the stable isotopes in the compartments (bulk soil carbon, and individual monomers of hydrolysable lipids) of the disturbance control plots as natural abundance values, and $^{13}C/^{12}C_{labeled}$ is the atomic ratio in the corresponding compartments in the plots with added labeled root litter.

As individual isotope values can vary a lot in between different homologues for each compound-class specifically in isotope labeling experiments, a more meaningful measure was chosen to express the $\delta^{13}C$ values of the respective compound-classes that can be assigned to the same carbon source (Wiesenberg et al., 2008). The $\delta^{13}C$ values and $^{13}C$-excess for the most abundant compound classes (*n*-alcohols, *n*-fatty acids, diacids, and $\omega$-hydroxy acids) within the hydrolysable lipid fractions were calculated separately as weighted means of individual compounds within each compound class. This means within each compound class, weightings will be given to each monomer in this compound class (e.g., $\omega$-hydroxy acids) based on the proportional contribution of individual monomer (e.g., $C_{24}$ $\omega$-hydroxy acids) to the total concentration of this compound class. Then for individual monomers, their weightings will be multiplied by their $\delta^{13}C$ values, and we sum up all the monomers identified to get the weighted mean $\delta^{13}C$ values for this compound class:

$$\mu_c = \sum_{i=a}^{b} (x_{ci} \times w_{ci}) \tag{5}$$

Where $\mu$ denotes the average value and subscript $c$ represents different compound classes, $x$ denotes the value of either $\delta^{13}C$ or $^{13}C$-excess, a and b represent the lower and upper limits of the respective carbon number range, $w_i$ indicates the relative abundance of the individual compounds within compound class $c$.

The amount of root carbon that was recovered in bulk soil was calculated by dividing the amount of $^{13}C$-labeled root-derived carbon left in the soil by the amount of carbon added with the original labeled roots. The proportion of root-derived carbon ($f_{root}$) was estimated by using a simple mixing model (Hicks Pries et al., 2018):

$$^{13}C\ atom\%_{sample} = {}^{13}C\ atom\%_{DC} \times f_{soil} + {}^{13}C\ atom\%_{root} \times f_{root} \tag{6}$$

$$f_{root} + f_{soil} = 1 \tag{7}$$

$$Recovery_{root} = \frac{f_{root} \times M_{soil} \times C\%_{soil}}{0.14 \times 0.463} \tag{8}$$

Where $^{13}C\ atom\%_{sample}$ is the $^{13}C$ atom% of soil samples where labeled root litter was added; $^{13}C\ atom\%_{DC}$ is the $^{13}C$ atom% of soil samples in the disturbance control plots; $^{13}C$ atom%$_{root}$ is the $^{13}C$ atom% of initial $^{13}C$ labeled roots; $f_{soil}$ and $f_{root}$ are the proportion of carbon

originally derived from native soil and labeled root litter, respectively. $M_{soil}$ is the mass of the soil sample and $C\%_{soil}$ is the carbon content of the corresponding soil sample. 0.14 is the mass of root litter (g) added at individual soil depth and 0.463 is the carbon content of the added root litter.

The decay rate, $k$, of initially added $^{13}C$-labeled roots and root-derived hydrolysable lipids was calculated based on the following model (Olson, 1963):

$$-kt = ln\,\frac{M_t}{M_0} \tag{9}$$

where $M_0$ is the mass of original roots or root-derived hydrolysable lipids, $M_t$ denotes the mass of root-derived carbon or hydrolysable lipids at time $t$, and $t$ is the duration of incubation which is three years in our study. The model assumes that $M$ is a well-mixed carbon pool with first-order decay kinetics. The residence time is the reciprocal of $k$.

The priming effect of added $^{13}C$ labeled root litter was calculated using the mass of carbon in DC cores at individual depths as background values. Then, based on the calculations shown in 2.6.4, the proportion of native SOM that is left in the labeled cores within the same plot after three years of incubation was calculated:

$$Priming\ effect = \frac{MSOM_l - MSOM_{DC}}{MSOM_{DC}} \tag{10}$$

Where $MSOM_l$ and $MSOM_{DC}$ denote native SOM left in labeled cores and SOM in DC cores, respectively. The priming effect for hydrolysable lipids was calculated with the same approach. One outlier is excluded in priming effect calculation at mid-depth in ambient plots. It is noteworthy that this is only an estimated proxy used for priming effect. The more classical and straightforward way to calculate priming effect is described in (Schiedung et al., 2023).

**2.7 Statistical analysis**

All statistical analyses were performed in the RStudio interface 2024.12.1.563 (Posit team, 2025) with R version 4.4.2 (R Core Team, 2024). We used linear mixed effects models (LMEs) in the '*nlme*' package (v3.1.166; Pinheiro et al., 2024) to test the fixed effects on responsive variables. All the plots were created using the '*ggplot2*' package (v3.5.1; Wickham, 2016). Homoscedasticity and normality were visually checked by residual plots and Q-Q plots. If model assumptions were not met, the data was log-transformed.

Response of bulk soil organic carbon concentration was tested to warming, root treatment, depth, and their interaction. $^{13}C$-excess was tested in response to warming, depth, and their interaction. The hydrolysable lipids concentration normalized to soil organic carbon (SOC; mg g$^{-1}$ SOC) was tested in response to warming, root treatment, depth, and their

interaction. We tested warming, depth, and their interaction on hydrolysable lipids mass change, and $^{13}C$-exces of each compound class separately. To test whether chain length increases suberin-derived monomers resistance to decomposition, we used warming, depth, carbon number (refers to the total number of carbon atoms in the backbone of the respective suberin monomers), and their interactions as fixed effects to evaluate their effects on mass change of individual $\omega$-hydroxy acid and diacid monomers. For each statistical test, we regarded block as a random effect.

Model fits were evaluated by Akaike's Information Criterion (AIC; Akaike, 1998). We conducted a backward stepwise model selection to remove fixed effects that increased AIC values by 5 or more (Pegoraro et al., 2025; in review). The alpha level was set to $\alpha = 0.05$ as significant in all statistical tests and a $p$ between 0.05-0.1 was considered marginal.

## 3. Results

### 3.1 Soils without root litter incubation

In ambient plots, the normalized hydrolysable lipids concentration decreased from 18.4 $\pm$ 4.4 mg g$^{-1}$ SOC in topsoil to 11.0 $\pm$ 2.7 mg g$^{-1}$ SOC in the deep soil, whereas it decreased from 18.6 $\pm$ 3.8 mg g$^{-1}$ SOC in topsoil to 12.5 $\pm$ 2.1 mg g$^{-1}$ SOC in the deep soil in warmed plots (Fig. 1a). However, this average decrease of hydrolysable lipids concentration with depth was not significant ($p = 0.460$; Table S10). Warming did not significantly alter the normalized hydrolysable lipids concentration ($p = 0.472$; Table S10) across the soil profile.

In general, warming did not significantly alter the relative contribution of each compound class to the total hydrolysable lipids ($p > 0.1$; Table S21) and depth only significantly decreased proportion of mid-chain acids ($p = 0.003$; Table S21). In ambient topsoil, $\omega$-hydroxy acids accounted for the largest proportion of total hydrolysable lipids by 31 %, followed by fatty acids (27 %), mid-chain acids (21 %), diacids (13 %), while alcohols only make up 9 % of the total (Fig. 1; Table S22). At the same depth, warming did not significantly affect the relative contribution of each compound class ($p > 0.1$; Table S22). At mid-depth (45-49 cm), the proportion of each compound class to total hydrolysable lipids was consistent compared to topsoil although contribution of mid-chain acids marginally decreased by 7 % ($p = 0.072$; Table S22). In the deep soil (85-89 cm), the proportion of mid-chain acids further significantly decreased by 10 % compared to its contribution in topsoil ($p = 0.011$; Table S22). Albeit much higher average contribution of fatty acids (45 %) in ambient deep soil, it did not significantly differ relative to topsoil ($p = 0.360$; Table S22). At this depth, warming

did not significantly alter the proportions of all the compound classes. However, elevated temperature marginally decreased fatty acids and increased diacids proportion by 16 % ($p$ =

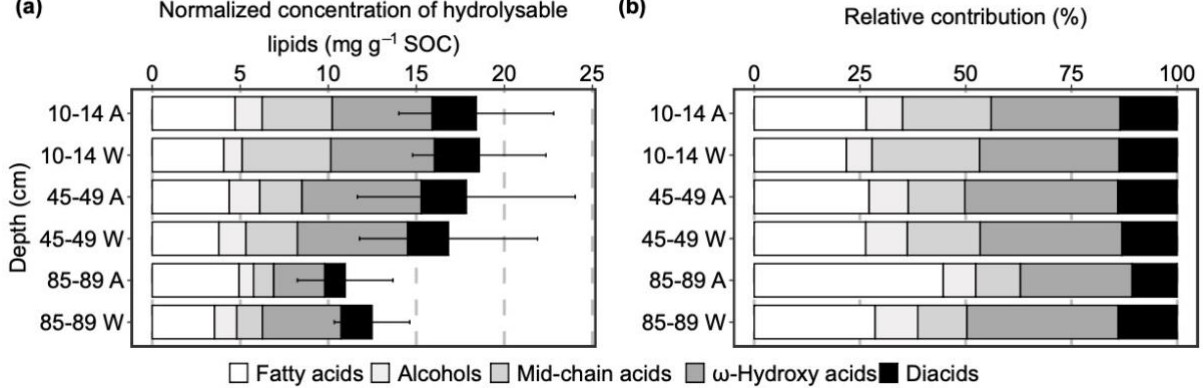

Figure 1: (a) Hydrolysable lipids normalized to soil organic carbon (SOC) concentrations (mg g$^{-1}$ SOC) in disturbance control (DC) with ambient temperature (A) and warming (W) in 2019 after 3 years *in-situ* incubation (mean ± SE, n = 3); (b) Proportions of individual compound classes to total hydrolysable lipids identified in DC cores (mean, n = 3). For clarity of the visual presentation SE error bars are shown cumulatively.

0.084; Table S23) and 3 % ($p$ = 0.064; Table S23), respectively.

### 3.2 Soils with root litter incubation

$^{13}$C-excess of bulk SOC did not significantly change with warming ($p$ = 0.527; Table S8) but increased significantly with depth ($p$ = 0.001; Table S8). At 10-14 cm, the $^{13}$C-excess of bulk SOC was on average 17.0 % lower in warmed than in ambient plots (Table S9), but this difference was not significant ($p$ = 0.527; Table S8). At 45-49 cm, $^{13}$C-excess of bulk SOC on average increased by 2.7 times from 3.8 % in topsoil to 10.5 % at this depth ($p$ = 0.016; Fig. 2; Table S9). In the deep soil, the $^{13}$C-excess on average furthered increased to 22.8 % (Fig. 2).

Similar to bulk SOC, the weighted $^{13}$C-excess of each compound class was not significantly altered by warming ($p$ > 0.05; table S12) but significantly increased with depth ($p$ < 0.05; Fig. 2b; Table S12). In topsoil, warming on average decreased weighted $^{13}$C-excess for all the compound classes between 3.2 % (diacids) and 8.5 % (alcohols) compared to ambient conditions, however, this decrease was not significant ($p$ > 0.1; Table S12). At 45-49 cm, warming non-significantly increased weighted $^{13}$C-excess for all the compound classes ($p$ > 0.1; Table S14), except for fatty acids, where warming on average decreased their weighted $^{13}$C-excess (p = 0.920; Table S14). In the deep soil, warming on average largely reduced the weighted $^{13}$C-excess of fatty acids (28.0 %; $p$ = 0.356; Table S14), $\omega$-hydroxy acids (17.1 %; $p$ = 0.728; Table S14), and diacids (8.2 %; $p$ = 0.760; Table S14), and slightly lower the

weighted $^{13}$C-excess of alcohols (0.9 %; $p$ = 0.989; Table S14) and mid-chain acids (3.0 %; $p$
= 0.925; Table S14), although none of them was significant. The weighted $^{13}$C-excess of fatty
acids increased significantly from 10-14 to 45-49 cm by 25.5 % ($p$ = 0.039; Table S13),
whereas the weighted $^{13}$C-excess all the other compound classes non-significantly ($p$ > 0.1;
Table S13) increased between 2.5 % (alcohols; Table S13) to 3.2 % (diacids; Table S13) at the
same depth. At 85-89 cm, the weighted $^{13}$C-excess of fatty acids, $\omega$-hydroxy acids, and diacids
significantly increased compared to the values at 10-14 cm by 62.2 % ($p$ = 0.004; Table S13),
62.9 % ($p$ = 0.033; Table S13), and 38.1 % ($p$ = 0.015; Table S13), respectively. The weighted
$^{13}$C-excess of alcohols and mid-chain acids marginally increased by 73.3 % and 36.7 %

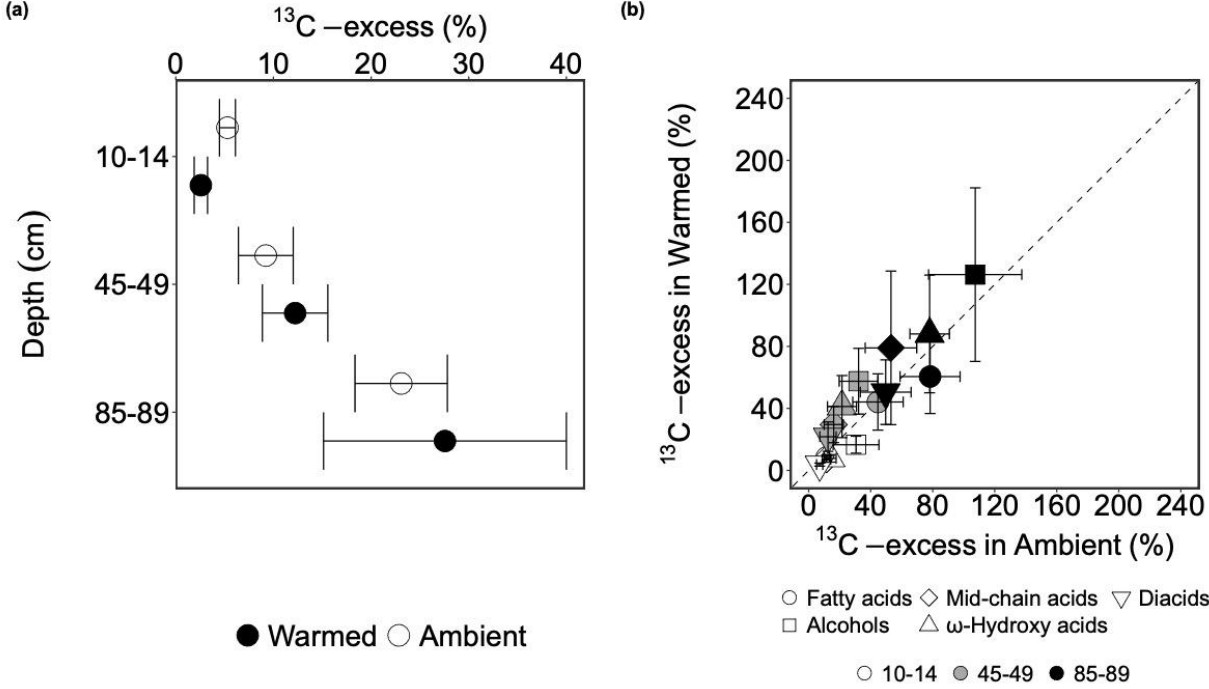

Figure 2: (a) The presence of carbon derived from the $^{13}$C-labeled root in the bulk soil after three years of incubation, expressed as $^{13}$C-excess of bulk soil carbon (Mean ± SE, n = 3) at 10-14, 45-49, 85-89 cm depth in 2019 in ambient plots (white circles) and warmed plots (black circles). (b) Comparing the means of weighted $^{13}$C-excess of each compound class from warmed plots (y-axis) and ambient plots at three depths (10-14, 45-49, 85-89 cm) with a 1:1 line (Mean ± SE, n = 3). Values above the 1:1 line indicate higher values in warmed plots than in ambient plots and values below the 1:1 line indicate the opposite.

correspondingly ($p$ = 0.054 and 0.051, respectively; Table S13) at the same depth.
The mass change of each compound class represents the proportion of hydrolysable
lipid-derived compound classes from the $^{13}$C-labeled root litter that remained after three years
of incubation. This metric allowed us to evaluate whether these compound classes experienced
a net loss or accumulation during the incubation. Values above and below 100 % indicates
accumulation and loss of corresponding compound classes, respectively.
The mass changes of different compound classes were affected differently by warming
and soil depth. Both warming ($p = 0.035$; Table S15) and depth ($p = 0.008$; Table S15) had
significant impacts on fatty acid mass change and the impacts of warming was independent
from depth ($p = 0.409$; Table S15). Fatty acids significantly accumulated across the soil profile,
resulting in an increased their mass change by 67 % ($p = 0.044$; Table S16) and 102 % ($p =$
$0.006$; Table S16) at mid-depth and in the deep soil, respectively. Although non-significantly
at 10-14 and 45-49 cm, warming lead to a consistent decreased fatty acid mass change across
the soil profile, particularly at 85-89 cm, where there was a significant lower (74 %; $p = 0.029$;
Table S17) recovered fatty acids with elevated temperature. The accumulation of fatty acids at
45-49 and 85-89 cm derived primarily from Hexadecanoic acid ($n$-$C_{16:0}$), Octadecanoic acid
($n$-$C_{18:0}$), and Octadecenoic acid ($n$-$C_{18:1}$), where the mass change of these monomers exceeded
110 % or even 300 % (Table S2), e.g. Octadecanoic acid ($n$-$C_{18:0}$) at 85-89 cm in ambient and
warmed plot ($348 \pm 49$ % and $181 \pm 24$ %; Table S2).
The effects of warming on the mass change of mid-chain acids were marginally depth-
dependent ($p = 0.057$; Table S15). In the topsoil, warming significantly decreased the mid-
chain acids recovered from hydrolysable lipids ($p = 0.029$; Table S17), whereas warming non-
significantly increased the mass change of this compound class at 45-49 ($p = 0.257$; Table S17)
and at 85-89 cm ($p = 0.729$; Table S17). Neither warming nor depth had significant effects on
the mass changes of all the other three compound classes ($p > 0.1$; Table S15). There was a
general trend that warming on average decreased the mass changes of alcohols, $\omega$-hydroxy
acids, and diacids in the topsoil, but increased them at mid-depth and in the deep soil (Table
S17), however, none of these changes were significant ($p > 0.1$; Table S15).
For fatty acids with a carbon number $\geq 20$, we did not observe significant warming effects on
the mass change ($p = 0.255$; Table S18), but marginal depth effects ($p = 0.076$; Table S18).
Warming on average reduced the mass change of fatty acids by 12 % at 10-14 cm, 7 % at 45-
49 cm, and 16 % at 85-89 cm, but these effects were non-significant ($p = 0.260$, 0.501 and
0.141, respectively; Table S20). Similar to fatty acids, there was no significant effects of neither
warming nor depth on the long-chain $\omega$-hydroxy acids and diacids mass change, and warming
effects was not dependent on depth ($p > 0.1$; Table S18). At 10-14 cm, warming led to non-
significant decreases in the $\omega$-hydroxy acids and diacids mass change by 16 % and 22 % ($p =$
0.334 and 0.316, respectively; Table S20). In contrast, although warming on average enhanced
the mass change of the two compound classes at both 45-49 and 85-89 cm, none of the changes
were statistically significant ($p > 0.1$; Table S20).

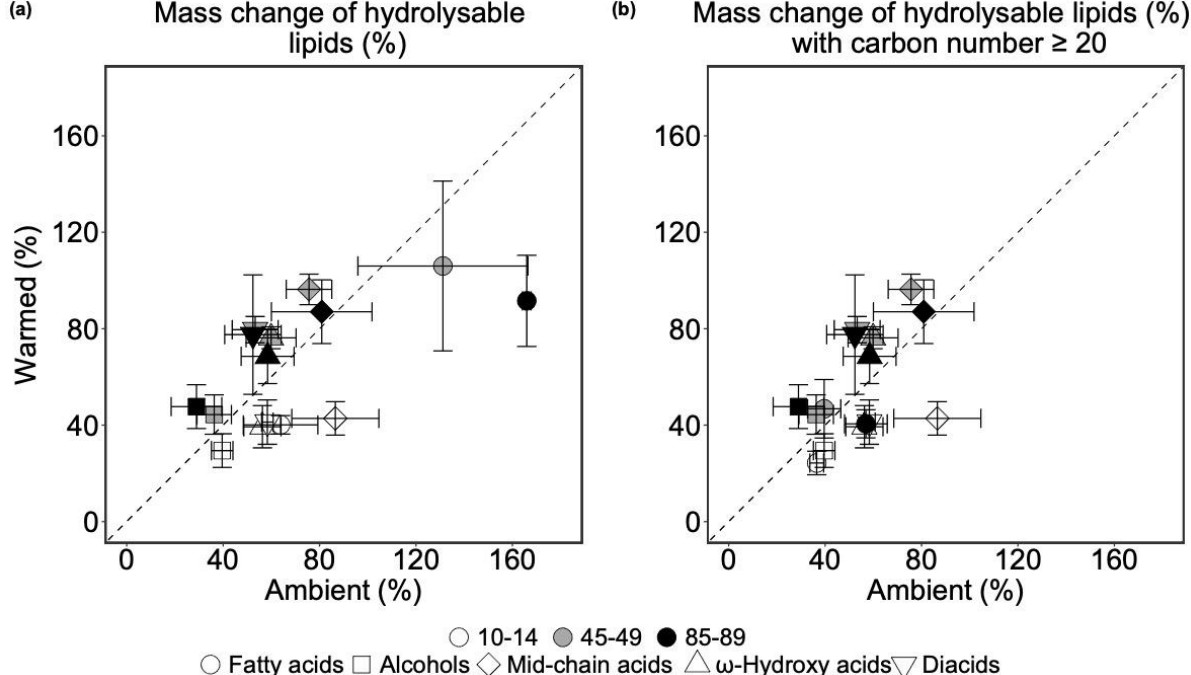

Figure 3: (a) Comparing mean of mass change (%) of each compound class in [13]C-labeled root-derived hydrolysable lipids (n = 3). Values above the 1:1 line indicate higher values in warmed plots than in ambient plots and values below the 1:1 line indicate the opposite. (b) Comparing mean of mass change (%) of each compound class in [13]C-labeled root-derived hydrolysable lipids (carbon number ≥ 20 for fatty acids, ω-hydroxy acids, and diacids; n = 3). In both plots, the error bars denote the standard error of each compound class in ambient and warmed plots in the individual soil depths. Values smaller than 100% indicate a loss and those larger than 100% indicate a gain of this compound class. Values above the 1:1 line indicate higher values in warmed plots than in ambient plots and values below the 1:1 line indicate the opposite.

## 3.3 Priming effect

471        Neither depth nor warming had significant impacts on the mass differences of pre-
existing SOC ($p = 0.558$ and $0.422$, respectively; Table S24). In topsoil, labeled root addition
led to SOC accumulation on average by 28 % in ambient plots, but this accumulation was not
significant ($p = 0.273$; Table S25). The effects of root addition decreased on average by 37 %
at mid-depth and 32 % at 85-89 cm, resulting in an on average loss of pre-existing SOC (Fig.
4a). However, these effects were non-significant ($p = 0.359$ and $0.375$, respectively; Table S25).
Warming caused an average decline of the bulk SOC mass change at 10-14 and 85-89 cm by
17 % and 14 % (Fig. 4a), and no-change at mid-depth, but none of them were significant ($p =$
$0.629$, $0.684$ and $0.991$ for 10-14, 85-89, and 45-49, respectively; Table S26).
The similar trend was observed for the mass change of pre-existing hydrolysable lipids as for
pre-existing SOC, but the direction was opposite in the topsoil and the deep soil (Fig. 4b).
Neither depth nor warming significantly altered the relative mass differences of pre-existing
hydrolysable lipids ($p = 0.310$, and 0.987 for depth and warming, respectively; Table S24). At
10-14 cm, labeled root addition led to a non-significant 19 % loss of pre-existing hydrolysable
lipids($p = 0.596$; Table S25). In the subsoil, the responses of relative hydrolysable mass change
were contrasting between 45-49 cm and 85-89 cm. At mid-depth, labeled root addition caused
a further non-significant decrease of the mass change of hydrolysable lipids by 35 % (Fig. 4b;
$p = 0.529$; Table S25), whereas in the deep soil, it caused a non-significant increase of the
hydrolysable lipids mass change by 45 % compared to topsoil (Fig. 4b; $p = 0.372$; Table S25).
In the topsoil, warming non-significantly decreased the mass change of hydrolysable lipids by
4 % ($p = 0.941$; Table S25). In the subsoil, warming had also contrasting effects on the mass
change of hydrolysable lipids (Fig. 4b) between mid-depth and deep soil. Warming on average
increased the mass change of hydrolysable lipids at 45-49 cm by 30 % and decreased it by 14 %
compared to corresponding ambient conditions, but none of them was significant ($p = 0.587$
and 0.784, respectively; Table S26).

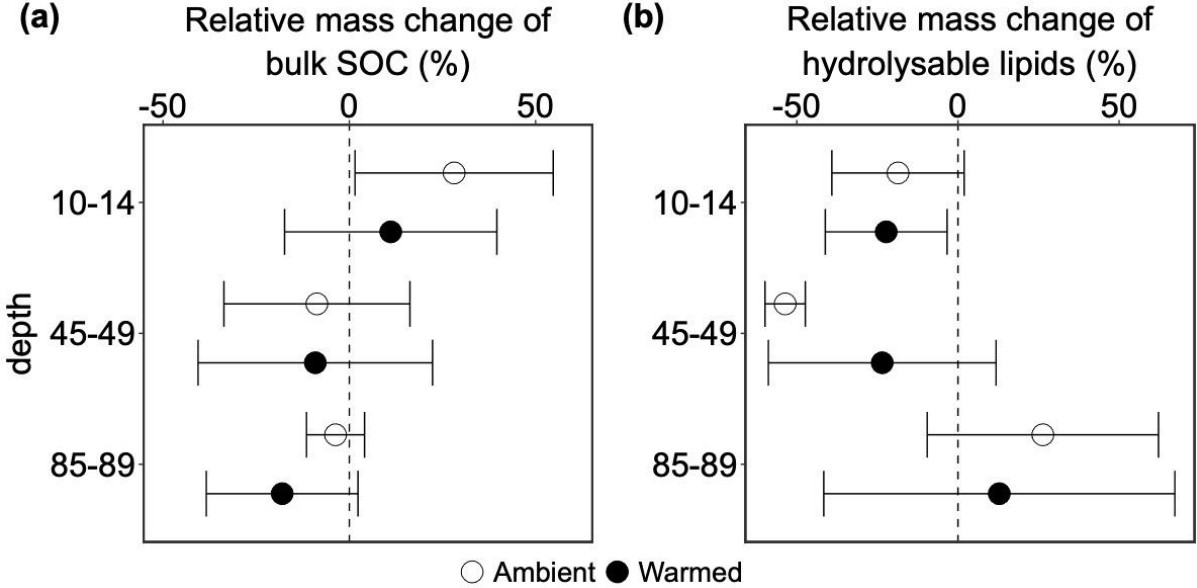

Figure 4: Relative mass difference of: (a) pre-existing bulk soil organic matter and (b) hydrolysable lipids between cores with and without addition of [13]C-labeled root litter (labeled – disturbance control cores). Negative values indicate the accelerated decomposition of pre-existing soil organic carbon (positive priming effect). Positive values indicate inhibition (negative priming) of decomposition (Mean ± SE, n = 3).

## 4. Discussion

### 4.1 Warming on hydrolysable lipids mass change was compound-dependent

#### 4.1.1 Warming significantly attenuated the accumulation of fatty acids in subsoil

The effects of soil warming on the decomposition of labeled root-derived hydrolysable lipids varied by compound class. Warming significantly decreased the mass change of fatty acids, independent of soil depth. In the topsoil (10-14 cm), although non-significantly, warming reduced mass change of fatty acids by 24 %, likely due to enhanced microbial decomposition. Similar acceleration of plant-derived input decomposition under warming has been reported at this site (Ofiti et al., 2021; Soong et al., 2021). In the topsoil, warming both increased carbon stocks and particulate organic matter, which is readily available substrates to microorganisms (Soong et al., 2021), and maintained microbial abundance while shifting microbial community structure toward taxa capable of degrading wider range of substrates (Dove et al., 2021; Zosso et al., 2021). Actinobacteria, known for breaking down complex carbon (DeAngelis et al., 2015; Goodfellow and Williams, 1983), increased in relative abundance under warming in our study (Pegoraro et al., 2025; in review).

In the subsoil (45-49, and 85-59 cm), fatty acid mass change exceeded 100 %, indicating accumulation of this compound class from other sources. This accumulation derived primarily from octadecanoic acid, octadecenoic acid, and hexadecanoic acid ($C_{18:0}$ fatty acids, $C_{18:1}$ fatty acids, and $C_{16:0}$ fatty acids, respectively; Table S2), which originate from both plant and microbial biomass. In contrast, long-chain fatty acids (carbon number $\geq$ 20), typically enriched in higher plant tissues (Harwood and Russell, 1984), did not accumulate. Therefore, the exceeding fatty acids originated likely from microorganisms, e.g. through phospholipid fatty acids synthesized during microbial growth (Joergensen, 2022; Zelles, 1997). [13]C-labeled substrate could be utilized by microorganisms and incorporated in microbial membrane lipids (Gunina et al., 2014; Yang et al., 2020). In our study, warming attenuated the accumulation of fatty acids in the subsoil, particularly at 85-89 cm, where there was a significant decline of the mass change of this compound by 74 %. This significant decrease implied a restricted decomposition of added root litter.

This pattern aligned with the concurrent observations from Pegoraro et al. (2025; in review), who observed higher phospholipid fatty acid concentrations and stronger [13]C-enrichment of fungal markers in ambient subsoil following root litter addition. Fungi preferentially utilized new plant-derived inputs (Lindahl et al., 2007) and incorporate these

substrates into their biomass (Williams et al., 2006). Warming reduced fungal relative in the subsoil (Pegoraro et al., 2025; in review), potentially slowing root litter decomposition and lowering fatty acid mass change. Additionally, warming reduced soil moisture by 18 % at mid-depth and 32% in the deep soil horizon in our study (Pegoraro et al., 2025; in review), which may have limited substrate and enzyme mobility, consequently restricting microbial access to fresh plan inputs, and thereby microbial decomposition (Manzoni et al., 2012b; Védère et al., 2020).

**4.1.2 No significant change of suberin markers in subsoil with warming**

In contrast to fatty acids, warming had no significant influence on the decomposition of alcohols, mid-chain acids, $\omega$-hydroxy acids, and diacids, although the effect on mid-chain was marginally depth-dependent ($p = 0.057$; Table S15). This lack of warming effect persisted also when focusing on $\omega$-hydroxy acids, and diacids with carbon numbers $\geq 20$ (Table S20), which are typically identified as suberin markers (Angst et al., 2016b; Franke et al., 2005; Mendez-Millan et al., 2011). These results suggested that warming did not alter the decomposition of added [13]C-labeled root litter across the soil depths after three years of incubation (Pegoraro et al., 2025; in review).

Prior studies have reported contrasting hydrolysable lipid responses to warming. At the temperate forest site studied here, Zosso et al. (2023) have discovered a significant loss of hydrolysable lipids by 28 % in the subsoil after 4.5 years of warming, attributed primarily to increased fine root mortality caused by warming (Ofiti et al., 2021). The discrepancy between our findings may arise from methodological differences that our approach involved mixing [13]C-labeled root litter into the soil, which may have reduced the spatial accessibility of microorganisms to fresh substrates, an effect potentially further amplified by the warming-induced declines in subsoil microbial abundance (Zosso et al., 2021). In another study in a hardwood forest, 14 months of warming did not affect suberin markers (Feng et al., 2008), whereas in a mixed hardwood forest, warming significantly decreased suberin concentration after 4 years of warming (Pisani et al., 2015), however, this pattern was not significant in the long-term (vandenEnden et al., 2021). Divergent results indicate that the responses of hydrolysable lipids to soil warming are dependent on various factors, e.g. experimental methods, vegetation, and duration of the experiment.

The unchanged suberin markers, coupled with significantly lower recovery of fatty acids in warmed relative to ambient subsoil, indicates a potential shift in SOC processing

pathways at these depths. In general, microbial carbon use efficiency (CUE) declined with soil depth (Pei et al., 2025; Spohn et al., 2016), a pattern also observed at our study site (Dove et al., 2021). Yet, the effect of warming on CUE remains uncertain (Zhang et al., 2022). If warming reduced microbial CUE in our incubation experiment, as observed elsewhere (Li et al., 2019; Li et al., 2024; Manzoni et al., 2012a), this could partly explain the greater fatty acid accumulation in the ambient subsoil. A decline in CUE under warming would favor microbial respiration over growth, thereby reducing the incorporation of substrates into microbial biomass (Manzoni et al., 2012a). Given the paucity of warming studies examining subsoil CUE, and its pivotal role in carbon storage (Tao et al., 2023), targeted experiments are needed to test this hypothesis in the future.

**4.2 Suberin markers' resistance to decomposition increased with chain length**

To further explore other factors governing the decomposition of suberin markers in our study, we examined the effects of chain length (number of carbon atoms in each monomer) on the mass change of $\omega$-hydroxy acids and diacids after three years of incubation. Chain length had a significant effect on the mass change of suberin markers ($p < 0.001$; Table S27) that they decomposed more slowly with longer chain length. This pattern is independent of warming, depth, and compound class. Our results are in line with multiple previous studies, which also reported decreased decomposability of hydrolysable lipids with increasing chain length by examining either single compound class (Kashi et al., 2023), or suberin- and/or cutin-derived polymers (Altmann et al., 2021; Angst et al., 2016a; Suseela et al., 2017). Generally, molecules with a low oxidation state or low O/C ratio (i.e., more reduced compounds such as lipids) provide less favorable energy yields for microbial oxidation, which often translates into slower decomposition rates (Kleber, 2010; LaRowe and Van Cappellen, 2011). As chain length increases, the carbon atoms in respective compound classes exhibit progressively lower oxidation states (Chakrawal et al., 2020). Therefore, the microbial utilization of long-chain lipids requires more energy investment than that of mid-chain lipids (Schönfeld and Wojtczak, 2016), which may explain the slower decomposition of long-chain $\omega$-hydroxy acid and diacid observed in our study.

It is noteworthy that when chain length was tested together with warming and depth on the mass change of $\omega$-hydroxy acids and diacids monomers, warming emerged as a significant factor ($p < 0.001$; Table S27), and its effect was depth-dependent ($p < 0.001$; Table S27). Specifically, warming significantly enhanced the loss of suberin monomers in the topsoil ($p =$

0.002; Table S29), but reduced it at 45–49 cm and 85–89 cm ($p < 0.001$ and $p = 0.008$,
respectively; Table S29). This pattern was also observed at compound class level, although
with limited statistical power due to limited number of replicates. The stronger statistical
significance at the monomer level reflects the greater number of observations (i.e., higher
degree of freedom) and lower variability within individual monomers compared to the
aggregated variability among monomers when grouped at the compound class level. Therefore,
our results can confirm that the effect of warming on suberin marker decomposition was depth-
dependent.

600       Root-derived hydrolysable lipids, especially those suberin markers, degraded more
slowly than bulk root carbon, indicated by higher proportions of hydrolysable lipids remaining
in the soil than the bulk root recovery (Pegoraro et al., 2025; in review). The depth-dependent
effects of warming lead to depth-specific mean residence time (MRT) of these suberin markers
(chain length $\geq 20$). In the topsoil, warming reduced the MRT of $\omega$-hydroxy acids and diacids
due to accelerated decomposition from 5.8 and 5.6 years in ambient plots to 3.4 years and 3.6
years. In the subsoil, slower decomposition of suberin markers caused by warming enhanced
MRT compared to ambient conditions, particularly at 45-49 cm with warming, where the MRT
for of $\omega$-hydroxy acids and diacids was 11.9 and 15.0 years, respectively. The suberin markers
had a shorter MRT in our study compared to a previous study, where Feng et al. (2010) reported
a decadal MRT of hydrolysable lipids between 32 to 34 years. This discrepancy could be related
to the fact that our model substrates are grass roots (*Avena fatua)* with less lignin and lower
C/N ratios than local woody roots (Hicks Pries et al., 2018; Silver and Miya, 2001).
Hydrolysable lipids without association to lignin could be steadily decomposed (Angst et al.,
2016a). Additionally, an underestimation of MRT could exist in our study since we have a
much shorter experimental duration compared (Feng et al., 2010).

616       In prior study, Suseela et al. (2017) reported that elevated temperature and $CO_2$
increased the chain length of $\omega$-hydroxy acids in grass roots (*Boutelou gracilis*). Together with
our finding that the decomposability of suberin markers decreases with chain length, this raises
the possibility that plant physiological adaptations to future warming could enhance the
persistence of root compartments against microbial decomposition and consequently mitigate
potential soil carbon losses under warming.

**4.3 No priming effect of added $^{13}C$-labeled root litter after three years of incubation**

Fresh biomass input can stimulate (prime) the decomposition of native, pre-existing SOM. This is especially relevant in subsoils, where SOM might have existed for a long time, from decades to millennia (Fontaine et al., 2007; Luo et al., 2019; Shahzad et al., 2018). Such priming could offset long-term carbon sequestration, especially in subsoil where there is usually substrate limitation (Bernard et al., 2022; Bingeman et al., 1953).

Three years after adding $^{13}C$-labeled root at three different soil depths, we found no evidence for significant priming across the whole soil profile, either for pre-existing bulk carbon or hydrolysable lipids. This likely reflects the fact that fresh input was added only at the beginning of the incubation. Priming is a transient response to fresh carbon input (Schiedung et al., 2023) and is commonly strongest at the beginning of the incubation (Fontaine et al., 2007; Tao et al., 2024), a pattern that has been also reported in lipid decomposition experiments (Angst et al., 2016a; Kashi et al., 2023). Therefore, any initial effect may have dissipated in the long-term. In addition, our calculations of priming effects showed large error bars, likely due to heterogeneous background SOC concentrations, particularly in the subsoil. For example, the mass of SOC in root litter added to the topsoil (10-14 cm) represented only 4.0-8.5% of pre-existing SOC, whereas at 85-89 cm, the same litter addition corresponded to 10.5-39.3% of pre-existing SOC. This wide range in the proportion of added to native SOC, alongside reduced microbial abundance under warming may have biased our estimates and masked potential priming effects. Therefore, although our back-of-the-envelope calculation provides a preliminary estimate, the strength of the interpretation is limited. Short-term monitoring of $CO_2$ fluxes and their partitioning between $^{13}C$-labeled root- and native SOC- derived, would offer more robust insights into whether root litter addition primes native SOC decomposition, particularly in the subsoil.

**5. Conclusion**

Our experiment is among the first *in-situ,* long-term incubations to investigate the effect of warming on the decomposition of simulated new substrates input across different soil horizons under natural field conditions. Using compound-specific isotope analysis with specific focus on root-derived molecular markers, we traced their fates through microbial mineralization, assimilation and/or stabilization via e.g. aggregate occlusion, and organo-mineral association (Pegoraro et al., 2025; in review). Distinct responses among compound classes across soil depths and temperature treatments improved our understanding of how

warming shifts the fate of root-derived molecular markers. Therefore, we recommend applying
this technique or more advanced variants, such as position-specific $^{13}$C-labeling, in the future
to better understand soil carbon dynamics under climate change. Position-specific $^{13}$C-labeling
technique can pinpoint which substrates or carbon functional groups are more rapidly
mineralized, incorporated into microbial biomass, or associated with minerals (Apostel et al.,
2015; Kashi et al., 2023; Schink et al., 2021).

660       Three years of incubation enabled us to track the fate of $^{13}$C-labeled root litter and assess

long-term dynamics of relatively stable compounds like hydrolysable lipids in the soil.
Collectively, at the compound class level, hydrolysable lipids showed compound class-
dependent responses to warming. Warming had the most pronounced effects on fatty acids,
consistently decreasing their mass change across the soil profile. However, the mechanisms
were fundamentally different between top- and subsoil. In the topsoil, enhanced microbial
decomposition accelerated fatty acid losses. In the subsoil, where there was accumulation of
fatty acids, warming attenuated this accumulation, particularly at depth, coincident with lower
microbial abundance and constrained hydrolysable lipids decomposition due to low soil
moisture. At the monomer level, suberin markers, $\omega$-hydroxy acids and diacids, were more
resistant to decomposition than bulk root-derived carbon, with their resistance increasing with
chain length. Moreover, warming exhibited depth-dependent effects on the decomposition of
suberin-derived monomers–significantly accelerated their decomposition in the topsoil while
suppressing it in the subsoil.
Substantial spatial heterogeneity was observed, especially in the subsoil, and this heterogeneity
was partially reflected by wide range of added carbon to background SOC ratios. The
disproportionate carbon inputs across depths may have biased our results, particularly in
subsoil where higher soil moisture and microbial abundance under ambient conditions favored
microbial decomposition. Along with the limited number of observations in our study, the
capability to generalize our findings is constrained. Therefore, future *in-situ* warming
experiments with greater emphasis on subsoil processes and enhanced replications are needed
corroborate our findings.**Code/Data availability**
The data used in this study are available from the ESS-DIVE repository.

**Competing interests**
The authors declare that they have no conflict of interest.

## Author contributions

BS conducted lipids analysis and data interpretation, contributed to writing original draft, statistical analysis, and editing. EP shared resources and contributed to statistical analysis, writing review, and editing. MST applied for the funding, designed and maintained the warming experiment, contributed to statistical analysis, and writing review. CUZ conducted elemental analysis, introduced lipid analysis to me, and contributed to conceptualization, writing review and editing. GLBW supervised BS through the lipid analysis, contributed to methodology, conceptualization, data interpretation and validation, writing review, and editing. MWIS applied for funding and conceived DEEP C project, contributed to conceptualization, data interpretation, writing review, and editing.

## Acknowledgement

We thank Nicholas Ofiti, Tatjana Speckert for introduction and help of lipid analysis methods, Thomas Keller, Barbara Siegfried and Yves Brügger for lab support.

This study was supported by the Swiss National Science Foundation (SNF) as the DEEP C project (200021_172744) and the Belowground Biogeochemistry Scientific Focus Area by the U.S. Department of Energy, Office of Science, Office of Biological and Environmental Research, Environmental System Science Program, under Contract Number DE-AC02-05CH11231.

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
