# Peer review of "Warming accelerates the decomposition of root-derived 1 hydrolysable lipids in a temperate forest and is depth- and 2 compound class-dependent 3 Binyan Sun1, Cyrill Zosso1,3, Guido L. B. Wiesenberg1, 4 5 Elaine Pegoraro2, Margaret S. Torn2, a"

_EGUsphere, 2025_

## Author Response (AR1)

**RC1:**

**General Comments:**

**1. This study investigates the effects of whole soil warming on the decomposition of root litter by depth in the soil profile in Blodgett Experimental Forest. The authors found, after 3 years of +4 degree C warming, distinct depth-dependent decomposition of labeled root litter where warming accelerated the topsoil root litter, but not the subsoil root litter. This is an interesting and novel contribution to the field.**

Thanks for your constructive, critical comments, and thorough review of our manuscript. Based on the comments, we have re-written the result section in the way that we clearly report whether there is warming, depth, or their interaction effects with p-values. We increased the font size of the previous figures and change the style of fig. 1 that a direct comparison between warmed and ambient plots at each depth for normalized concentration and relative contribution of hydrolysable lipids is possible. Because the root-derived carbon data was reported in our sibling paper (Pegoraro et al., 2025; in review), we retrieved the figure reporting this result, changed the title with a more specific focus on hydrolysable lipids. Thus, we also re-wrote 4.1 in the way that we reported the responses of compound classes to warming. Additionally, we added the analysis of chain length of suberin-derived monomers on their resistance to decomposition.

**2. The introduction and results are clear and well-structured, but the discussion is very long.**

Thanks, and we shortened the discussion and removed the repetitive text.

**3. Please acknowledge the limitations in study design regarding the large spatial heterogeneity of soil properties (plus microbial community differences and conditions at depth) at this site and the low sample size of n=3. C inputs (amount and type) and the effects of warming are expected to differ by depth in the soil profile, yet the study design does not account for this. Since there was increasing variability in microbial communities with depth, this should be more directly addressed since the main conclusion of this work is the depth-dependent responses.**

We agree, soils in general have a large spatial heterogeneity, in our case especially in subsoils. This high spatial heterogeneity is shown in our study with large standard errors and further reflected by the large standard errors of subsoil PLFA analysis reported by our sibling paper (Pegoraro et al., 2025, in review). We mentioned this problem specifically in discussion at line 593 and in the conclusion from line 670 to 672.

**4. The use of a single root litter type from an annual grass is also a limitation, as this coniferous forest can be expected to have root contributions from fine roots of conifers which should have different chemical composition like lignin and lipids. Please justify the use of wild oat roots instead of conifer roots.**

We completely agree that coniferous tree fine roots could have been a better substrate for such an *in-situ* decomposition experiment. Uniform [13]C-labelling of slowly growing trees is technically extremely challenging but can be achieved with fast growing wild oat. Therefore we used wild oat as a model substrate, also in several fine root decomposition experiments

(Castanha et al., 2018; Hicks Pries et al., 2018). This is specifically stated in the revised manuscript between line 162 and 165

**5. If the natural temperature gradient was maintained, there may still be artifacts of the heating coil that are not accounted for, like differentials in soil drying by depth and consequent influence on decomposition dynamics at different depths. There could also be differences in microbial community distribution in close proximity to the heating coils vs. further away.**

This argument is absolutely right. The soil moisture was monitored throughout the whole period of experiment across soil profile and its natural gradient was kept (Pegoraro et al., 2025, in review, Hicks Pries et al. (2017). The incubation experiment was conducted with a consistent distance to the heating rods in all the plots, and the warming magnitude was maintained. Therefore, we considered the effects of the distance of heating rods to where we conducted our incubation experiment was not significant.

**6. Measuring after 3 years does not capture short-term priming effects, which would be expected more immediately than 3 years later. Please take this into account when addressing priming in the results and discussion sections. For example, the statement on L439, "Thus, positive priming occurred," from my perspective, cannot be so definitive.**

Agreed. We removed this statement and modified the section 4.3 and discussed the limitations of our way to calculate priming effects.

**7. Regarding statistical analysis and model selection, LME and AIC were used to assess the best fit models but it is not directly stated which models were compared. This could be added in a supplementary section. It is also unclear whether the depth, temperature, and their interaction were modeled as fixed or random effects in all cases.**

All the model structure and their results are not reported in the supplementary material.

**8. The study suggests that microbial activity was higher in ambient subsoil compared to warmed subsoil, based on the accumulation of mid-length fatty acids, but if microbial activity was higher in ambient conditions, one would expect greater decomposition of SOM and added root material. Instead, results suggest greater root-litter preservation in ambient plots. Perhaps there are alternative explanations for the accumulation of fatty acids, such as selective preservation, microbial necromass accumulation, or sorption to mineral surfaces.**

Agreed and thanks for the suggestions. We wanted to emphasize that the accumulation of mid-length fatty acids was a sign of added labeled root litter being more incorporated in microbial biomass under ambient conditions. This was also discovered in Pegoraro et al. (2025; in review), where they found significant higher microbial abundance and $^{13}C$-enrichement in microbial biomass in ambient subsoil. We did find almost 50 % of labeled root-derived $^{13}C$ protected by aggregate occlusion and organo-mineral association after three years of warming (Pegoraro et al., 2025; in review), but this did not differ between treatment and depth. And we did not conduct hydrolysable lipids on each soil density fractions, therefore, we cannot give a more definitive answer. We avoid using the term "microbial activity" because this was not directly measured by our study.

**9. While bulk root-litter decomposition was not significantly different between warmed and ambient plots, lipid composition changed, and fatty acid accumulation occurred under ambient conditions. The authors state that subsoil decomposition was unaffected**

**by warming, but this contradicts their molecular marker results showing that microbial metabolism and decomposition pathways did shift.**

In the revised manuscript, we specifically reported that at compound class level, warming only significantly affected the responses of fatty acid and we attributed this to the accumulation of mid-chain fatty acids occurred because of labeled root litter being incorporated in microbial biomass as suggested by PLFA data (Pegoraro et al. 2025; in review). Although warming had consistent effects on other compound classes (accelerated their decomposition in the topsoil and slowed it in the subsoil), this pattern only had limited statistical power, possibly due to limited number of replicates. However, when adding statistical analysis on the effects of chain length on the resistance of individual suberin monomers, we found this depth-dependent warming effects significant, perhaps due to larger number of observations. And we discussed a potential decomposition pathways shift between line 557 and 568, but this is only a speculation which needs to be further confirmed by future studies.

**Specific Comments:**

**10. L76-77: Unclear what is meant by "harnessing roots"**

We added the explanation of "harnessing roots" at line 117.

**11. L113-114: It is unclear if soil depths were heated the same amount or not from this sentence. I recommend explaining what the natural temperature gradient is rather than refer to another paper.**

We added the explanation between line 155 and 158.

**12. L121-146: The coring system, excavation and root additions are a little confusing. For each depth, was a soil core extracted, then the labelled roots added to the hole, then the soil replaced for that depth? Or was the soil inside the core, plus the core itself, left in the hole for the duration of the experiment?**

Sorry for the confusion. We added more detailed explanation in section 2.2.

Briefly, the soil core was added back to the drilled hole for the duration of the experiment. The customized soil core (polycarbonate) was hammered together with an aluminum tube outside of the soil core. The soil core contained 4 parts (0-10, 10-45, 45-85, and 85-95 cm). The soils in the latter three soil core parts were removed and mixed with labeled root and then added back to the corresponding part. Then all the parts were threaded on each other and added back to the soil.

**13. L148-150: Why were those specific depths chosen? Is it because of the known rooting depths of conifers in that forest? If so, this should be described in the site description section at the top of methods.**

We added this between line 196 and 199 in section 2.2.

**14. L222: Specify what is meant by region-specific in this context.**

We added definition at line 275.

**15. L322: The figure text is very small and hard to read. Instead of separating the ambient and warmed graphs, it would be better to have the sets of bars next to each other for**

direct comparison (e.g., ambient and warmed 10-14 cm, ambient and warmed 45-49, etc.).

We already increased the font size and group the depth.

**16. L339: Line about the error bars is not needed in the text since it's in the figure captions.** We deleted.

**17. L345: Was this difference statistically significant? Specify either way.**

Now we reported data with p-values.

**18. L467-468: For clarity, instead of "This argues for co-metabolic decomposition of the added root litter," "this indicates…"**

We removed this sentence to avoid confusion.

**19. L642-644: What is meant by "the warming was heterogeneous" in this sentence?**

We wanted to say the warming effects on root-litter decomposition were heterogenous in the subsoil mainly due to its spatial heterogeneity, much less substrate, and much lower microbial abundance. We already modified that.

**Technical Corrections:**

**20. L46: missing word: "…biotic factors THAT could change…"**

We modified as suggested.

**21. L50: grammatical errors: "Moreover, roots impact on SOM dynamics in subsoil in two way:"**

We re-phrased the sentence to make it clearer.

**22. L50-52: What is meant by "They are more likely to form stable SOM to aboveground plant biomass"?**

We re-phrased the sentence to make it clearer.

**23. L54-55: Grammar revisions needed.**

We re-phrased the sentence.

**24. L68-69: "Besides" is an awkward way to start a sentence.**

We re-phrased the sentence.

**25. L 70: grammatical errors**

We re-.phrased the sentence.

**26. L71: Missing the word "the"**

We modified as suggested

**27. L96-97: Revise second hypothesis for clarity and maintain consistency in tense used. Relative accumulation to what?**

We did the revision as suggested.

**28. L176: Write out the word dichloromethane for clarity and consistency with other acronyms.**

We did the revision as suggested.

**29. L188: Remove extra space after min**

We checked all the extra space and removed it.

**RC 2:**

**General Comments:**

This study employed an innovative in situ whole-soil profile warming experiment combined with $^{13}$C-labeled root litter to systematically investigate the response of root-derived carbon decomposition to climate warming across different soil depths in a temperate forest. The experimental design is notably novel, and the application of molecular markers (hydrolysable lipid monomer analysis) provided high-resolution data on the chemical transformation of root carbon. Results revealed that warming significantly accelerated the decomposition of root carbon in surface soils (10–14 cm), while having no significant effect in subsoils (45–89 cm), highlighting a pronounced depth-dependent heterogeneity in soil carbon turnover. Moreover, the accumulation of long-chain ω-hydroxy acids and dicarboxylic acids in subsoils suggests that warming may retard the decomposition of recalcitrant carbon by reducing microbial activity or altering substrate availability. The study integrated $^{13}$C-excess isotopic tracing with stoichiometric analysis to robustly verify carbon fate from multiple perspectives. The data are comprehensive and the methodology is rigorous, providing critical insights into the depth-dependent responses of soil carbon cycling under climate warming. Nevertheless, certain aspects of the analytical methods, interpretation of results, and experimental design details require further clarification or refinement to enhance the reliability and robustness of the conclusions.

**Specific Comments:**

1. **The low sample size of only n=3 and the large variation of the subsoil data (such as the extremely wide $^{13}$C-excess error bar of 85-89 cmin Figure 2) may mask the true effect of warming. It is suggested to explain the statistical power (such as post-hoc power analysis), or discuss the influence of small samples on the conclusion.**

   Thank you for the detailed review and valuable feedback on our manuscript. We added the results of linear mixed effects models in the supplementary material with post-hoc test. We mentioned the problem of small sample size specifically in discussion at line 593 and in the conclusion from line 670 to 672

2. **This study used the root systems of annual grasses (wild oats) instead of those of local dominant coniferous trees (such as pine trees). The lignin content of wild oats is low and the C/N ratio is low. The decomposition rate may be faster than that of woody roots, which may overestimate the effect of warming on the carbon loss of topsoil. It is suggested to discuss this limitation or supplement the control experiments on coniferous tree roots.**

   We completely agree that coniferous tree fine roots could have been a better substrate for such an *in-situ* decomposition experiment. Uniform $^{13}$C-labelling of slowly growing trees is technically extremely challenging but can be achieved with fast growing wild oat. Therefore we used wild oat as a model substrate, also in several fine root decomposition experiments (Castanha et al., 2018; Hicks Pries et al., 2018). This is specifically stated in the revised manuscript between line 162 and 165

3. **Heating cables may cause soil moisture gradients (such as subsoil drying), but the influence of temperature increase on the moisture content of each soil layer is not quantified in the text (Line 116). It is suggested to supplement the monitoring data of soil temperature and humidity or discuss the possible impact of heating on the habitat of microorganisms.**

   Thanks for the constructive comment. The soil moisture was monitored in the duration of the experiment. The data is available in our sibling paper (Pegoraro et al., 2025; inreview) and the impact of changing soil moisture on microbial biomass (PLFA) was also being discussed in that paper. A specific discussion of soil moisture on microbial decomposition is discussed in this manuscript between line 529 and 533.

4. **The paper mentions the selection using the Linear Mixed Effects Model (LME) and the AIC model, but does not clearly state the specific structures of fixed effects (such as warming and depth) and random effects.**

   We added the model structures in the supplementary material.

5. **It is claimed that "there is no significant primingeffect" (Line 439), but Figure 4a shows that there is negative excitation in the subsoil (inhibiting the decomposition of primary carbon). It is necessary to clarify the statistical test results ($p$ value), or modify the expression.**

   Sorry for the confusion. We added the p-values and was more cautious for our interpretation. A further discussion of the limitation of our methods to calculate priming effects was also discussed in the section 4.3.

6. **Warming in the subsoil did not change the total carbon content of the root system ($^{13}C$ recovery rate), but molecular markers indicated changes in microbial metabolism (such as fatty acid accumulation). It might be due to the increased input of microbial residues (PLFA contribution), or the enhanced physical protection of subsoil carbon (such as mineral binding) caused by warming. It is suggested to discuss the impact of changes in community structure in combination with the existing microbial data.**

   We added this PLFA discussion and its potential influence on the decomposition root-derived hydrolysable lipids between line 523 and 529. More detailed discussion of PLFA data please refer to Pegoraro et al. (2025; in review).

7. **The enrichment of C16-C18 fatty acids in the subsoil (>100% initial amount) may result from the input of microbial membrane lipids, but the interference from plant sources has not been ruled out. It is suggested to distinguish the contributions of microorganisms and plants through $\delta^{13}C$-PLFA analysis.**

   In Pegoraro et al. (2025): 1) root addition increased microbial biomass in ambient but not in warmed subsoils. 2) the very high $^{13}C$-excess of PLFA in all microbial groups identified (actinobacteria, fungi, Gram–, and Garm+) indicated that labeled root litter had been incorporated in microbial biomass. However, it is very difficult to distinguish the contribution of plant and microorganisms to mid-length fatty acids in our study, and this is why usually we do not consider these compounds as specific biomarkers both either suberin or PLFA. In the revised manuscript, there is a specific discussion of suberin monomers in section 4.2.

8.  **Figure 3 cannot visually compare the differences between ambientand warmed. It is suggested to change the presentation form of the chart.**

    We made the changes as suggested.

9.  **The results of the primingeffect in Figure 4 need to be marked with statistical significance (e.g. * _p_ < 05).**

    Actually, there was no statistical significance for priming effects data. We added p-values to each of the values reported in result session.

10. **Avoid overinterpretation (e.g. Line 439 "Thus, positive priming occurred").**

    We removed this overinterpretation .

11. **The 3-year experiment may have failed to capture the short-term excitation effect or the delayed response of the bottom carbon pool. It is suggested to discuss the necessity of long-term observation.**

    We discussed this at line 659 and 660.

**CC:**

This paper leveraged the Blodgett Forest whole-soil-profile warming experiment in a mixed coniferous temperate forest to examine warming effect on the decomposition of root-derived carbon, which serves as the primary organic inputs to soils. The study employed two approaches for this purpose by examining the lipid biomarkers of roots (suberin) in soils at three depths (10-14, 45-49, and 85-89 cm) after three years of warming treatment and via three-year *in-situ* incubation of 13C-labelled grass root-litter at each depth. The authors found that the decomposition of added root-litter was only accelerated in the topsoil (10-14 cm) but not in the subsoil (45-49 and 85-89 cm) with warming. Hence, the impact of warming on the decomposition of root-litter in a temperate forest is depth-dependent. Overall, the authors employed novel and complementary approaches to examine how root carbon decomposition respond to warming, which has significant implications for understanding how soil carbon cycling may be altered by climate change. The application of root-specific biomarkers and compound-specific 13C analysis in investigating root carbon turnover deserves applause. I have a few relatively minor comments/suggestions to improve the readability and to strengthen the conclusions of the paper.

We are grateful for your insightful comments and comprehensive review, which have greatly helped us improve the manuscript.

First, quite some of the results were not statistically significant (including the PE—I would say that no significant PE was induced). Please avoid overstating the results, which can be confusing. Some of the discussions are pure speculations without much data support. Please reduce these as well. I find some of the results repetitive, which can be made more succinct.

We re-wrote the result section with reported p-values without changing the main messages. Because the root-derived carbon recovery data was reported in our sibling paper (Pegoraro et al., 2025; in review), we retrieved the figure reporting this result, changed the title with a more specific focus on hydrolysable lipids. Thus, we also re-wrote 4.1 in the way that we reported the responses of different compound classes to warming. Additionally, we added the analysis of chain length of suberin-derived monomers on their resistance to decomposition. We also removed some repetitive parts to make it more succinct and avoided overstatement.

Second, as the authors mentioned, the dose of added roots was different on SOC basis for different soil layers. How would you expect it to influence the results? Can you specify? For instance, subsoil root decomposition may be underestimated due to the high dose of carbon added?

It's a very critical point. We discussed this briefly in the last paragraph of the conclusion.

Third, the application of root-specific biomarkers and compound-specific 13C analysis in investigating root carbon turnover deserves applause. How do you expect this approach to be used in the future? How would you recommend to improve its application? I would love to see the authors comment on this, which is a novel aspect of the study and worthy of further application.

Thanks for the comment. We added this discussion at the beginning of the conclusion.

**Additional detailed comments below:**

1. **Line 51: "than", not "to".**

   **We modified as suggested..**

2. **54: …debate continues, on how…**

   We re-phrased the sentence.

3. **96: The working hypotheses can be better refined. The second one is not really a hypothesis (it's known, right?).**

   We re-phrased the second hypothesis.

4. **How much did the examined lipids contribute to the added OC (with 13C labels)? Did the percentage change under warming?**

   That's a very good point. And we wanted to add this. However, there are several reasons that we did not include in the end: First, there are several monomers that contribute substantially to the total lipids. But because they co-elute together, it's hard to distinguish their $\delta^{13}C$ values (enrichment), and thereby identify and quantify them. Second, as discussed in the main text, accumulation of mid-chain fatty acids from other sources (other than hydrolysable lipids) could also bias this result. Therefore, for this manuscript, we decided not to include this data.

5. **505: more slowly.**

   We modified as suggested.

---

## Author Response (AR2)

**General comments: The author responses to reviewer comments and revisions are satisfactory. I commend the authors for their thorough and thoughtful revisions.**

**Specific comments:**

**L406: Suggest "Response of bulk soil organic carbon concentration to warming, root treatment, depth, and their interaction"**

Sincerely thanks for your thorough reviewing, and constructive suggestions to improve the quality of this manuscript.

I have changed this sentence to "Response of bulk soil organic carbon concentration was tested to warming, root treatment, depth, and their interaction" as suggested.

**L418: What is meant by "Carbon number"?**

I have added a brief explanation about what does carbon number mean after "carbon number" in the parentheses "refers to the total number of carbon atoms in the backbone of the respective suberin monomers"

**L446: What is the percentage in the parentheses?**

I deleted the percentage in the parentheses to avoid confusion.